# *Lactobacillus acidophilus* TW01 Mitigates PM_2.5_-Induced Lung Injury and Improves Gut Health in Mice

**DOI:** 10.3390/nu17050831

**Published:** 2025-02-27

**Authors:** Siou-Min Luo, Ming-Ju Chen

**Affiliations:** 1Department of Animal Science and Technology, National Taiwan University, Taipei 10617, Taiwan; siouminluo@ntu.edu.tw; 2Center for Biotechnology, National Taiwan University, Taipei 106038, Taiwan

**Keywords:** *Lactobacillus acidophilus*, intestinal protection, immune regulation, anti-PM_2.5_ injury

## Abstract

**Background/Objectives**: Exposure to fine particulate matter (PM_2.5_) causes significant respiratory and gastrointestinal health problems. In our prior research, we identified *Lactobacillus acidophilus* TW01 as a promising strain for mitigating oxidative damage, enhancing wound healing in intestinal epithelial cells, and protecting bronchial cells from cigarette smoke extract. Building upon these findings, this study examines the protective effects of this strain on lung damage induced by particulate matter (PM) through the gut–lung axis in mouse models. **Methods**: This study evaluated the protective effects of *L. acidophilus* TW01 against PM_2.5_-induced lung injury using two in vivo mouse models (OVA sensitization combined with PM_2.5_ exposure and DSS-induced colitis). **Results**: *L. acidophilus* TW01 exhibited significant protective effects in two in-vivo models, reducing pro-inflammatory cytokines (TNF-α, IL-6, and IL-5), modulating the immune response (IgG subtypes), and improving gut barrier integrity. Importantly, *L. acidophilus* TW01 increased the abundance of beneficial gut bacteria (*Bifidobacterium* and *Lactobacillus*). **Conclusions**: These findings highlight the significant protective/therapeutic potential of *L. acidophilus* TW01 in mitigating the adverse health effects of PM_2.5_ exposure, emphasizing the interplay between the gut and lung microbiomes in overall health. The multi-faceted protective effects of this probiotic suggest a novel, multi-pronged therapeutic strategy for addressing the widespread health consequences of air pollution.

## 1. Introduction

Fine particulate matter (PM_2.5_, with a diameter ≤ 2.5 µm) exposure leads to microbiota dysbiosis and metabolic disorders, potentially through the disruption of the lung–gut axis [1]. This manifests as lung oxidative stress, gut barrier dysfunction, and systemic inflammation, exacerbating inflammatory lung diseases like pulmonary emphysema, asthma, and lung cancer [2,3,4]. Exposure to PM_2.5_ triggers pulmonary inflammation and oxidative stress, contributing to a variety of respiratory ailments [2,5]. Oxidative stress has been identified as a principal mechanism of PM_2.5_ toxicity [6,7]. Additionally, PM-induced microbiota dysbiosis impairs gut barrier function and modulates serum lipopolysaccharide levels [8].

The impact of air pollutants extends to the gastrointestinal system, with PM_2.5_ affecting its structure and microbiota via respiratory exposure [9,10,11]. Long-term exposure is linked to ulcerative colitis [11,12,13] and elevated cancer risks, including lung, brain, breast, endocrine, prostate, and colorectal cancers [14]. PM_2.5_ alters gut microbiota composition and immune homeostasis [15,16,17].

The gut–lung axis describes the connection between respiratory and gut health, where the immune system, microbiota, and their metabolites play significant roles. Both the respiratory and gastrointestinal systems regulate inflammatory responses by modulating the secretion of chemokines and cytokines, which in turn influences immune system activity [16]. Chronic exposure to air pollution has been shown to alter the composition of the microbiota and affect microbial metabolite production, leading to pro-inflammatory responses and disrupted immune homeostasis [18]. Probiotics have emerged as beneficial interventions for pollution-induced health detriments, improving lung injury by restoring microbiota, enhancing gut barrier function, and rebalancing immune responses [15]. They offer protection against high PM_2.5_ levels [15] and mitigate cytokine overactivation and IgE production induced by PM_2.5_ [19].

In our previous research, we employed an in-vitro screening model that identified *Lactobacillus acidophilus* TW01 as a promising strain for mitigating oxidative damage, enhancing wound healing in intestinal epithelial cells, and protecting bronchial cells from cigarette smoke extract [20]. Building upon this foundation, the current study uniquely addresses the protective effects of *L. acidophilus* TW01 against PM-induced lung injury in a mouse model, specifically through the gut–lung axis. By utilizing two distinct groups of mice, we demonstrate not only its novel therapeutic potential but also the underlying mechanisms involved. This study fills critical gaps in our understanding of the gut–lung interaction and supports the development of targeted probiotic therapies for respiratory health.

## 2. Materials and Methods

### 2.1. Bacteria Preparation

*L. acidophilus* TW01 was anaerobically cultured in MRS broth (Difco Laboratories, Detroit, MI, USA) at 37 °C for 24 h until third passage, after which the bacteria were washed twice with 0.85% NaCl solution. Then, *L. acidophilus* TW01 samples were suspended in 10^7^ and 10^9^ CFU/mL with PBS buffer and stored in 4 °C for further research.

### 2.2. Animal Models

Six-week-old female BALB/cByJNarl mice were used for PM_2.5_ lung injury models and eight-week-old male C57BL/6 mice were used for the dextran sulfate sodium salt (DSS) colitis model. All mice were purchased from the National Animal Center and housed in a Specific Pathogen-Free (SPF) environment at 22 ± 2 °C with semi-humid conditions, maintaining a 12 h light/12 h dark cycle, at the National Taiwan University Animal Resource Center. All animal experiments were conducted in accordance with the relevant guidelines and legal requirements (PM_2.5_ certification number: NTU-111-EL-00134, approved on 31 March 2023, and DSS certification number: NTU-112-EL-00135, approved on 25 March 2023) and were approved by the Institutional Animal Care and Use Committee (IACUC) at National Taiwan University, Taipei, Taiwan.

### 2.3. PM_2.5_-Induced Lung Injury

The mice were randomly divided into four groups, each containing 10 mice, named Ctrl, NC, LD, and HD. After 7 days of pre-feeding with probiotics, all mice were sensitized with 20 μg OVA and 2 mg Al(OH)_3_ on Days 7, 14, and 21. On Day 14, we administered 1.8 mg/kg body weight of urban dust (SRM 1649b [21]; National Institute of Standards and Technology, U.S. Department of Commerce, Gaithersburg, MD, USA), which is a standardized reference material for particulate matter. The dust was suspended in 10 μL of phosphate-buffered saline (PBS) and delivered via anterior nasal cavity drops daily until Day 29. On Day 30, blood samples were collected and all mice were sacrificed. The simplified experimental flow chart is shown in Figure 1A.

#### 2.3.1. Bronchoalveolar Lavage Fluid (BALF) PM_2.5_-Induced Lung Injury Mice

After sacrificing the mice, 1 mL of PBS buffer was used to collect the BALF with 1 mL needles three times. The samples were centrifuged at 4 °C at 1200 rpm for 5 min. The first 1 mL of BALF was concentrated into 0.4 mL of PBS buffer for total cell counts. Cytokines were collected and analyzed from the BALF supernatant.

#### 2.3.2. Detection of Fibrosis Markers in PM_2.5_-Induced Lung Injury in Mice

Protein was extracted from the lung tissue of all PM_2.5_ mice and analyzed using Western blotting, which was performed with antibodies against phosphate-SMAD family member 3 (*p*-Smad3, Cell Signaling Technology, Danvers, MA, USA, C25A9, 1:1000), Smad3 (C67H9, 1:1000), TGF-β (Abcam, ab17969, 1:1000, Cambridge, UK), and GAPDH (Abcam, ab181602, 1:1000). Quantitative analysis of the Western blotting results was performed using ImageJ 1.54g, and the relative intensities of the target proteins compared to GAPDH are shown.

### 2.4. Dextran Sulfate Sodium Salt (DSS)—Colitis Treatment

All C57BL/6 mice were randomly separated into five groups, with 10 mice in each group: Ctrl, NC, P, TW01LD, and TW01HD. Ctrl served as the control group, NC as the DSS control group, and P as the commercial probiotic control, while the TW01LD group received 10^7^ CFU/mL of *L. acidophilus* TW01 and the TW01HD group received 10^9^ CFU/mL of *L. acidophilus* TW01. This method was described by Hung et al. [22]. The drinking water for the NC, P, TW01LD, and TW01HD groups was replaced with a 2.5% DSS solution for 7 days to induce colitis, following a 7-day pretreatment (molecular weight: 36,000–50,000 Da; MP Biomedicals, Aurora, OH, USA). Before DSS treatment, stool samples from all mice were collected to test for occult blood on Day 1 and Day 5. On Day 8, each mouse’s stool was tested using the stool occult blood test every day until the day before sacrifice. On the day of sacrifice, all mice were orally administered 200 μL of FITC-dextran (4000 Da, 46944, Sigma-Aldrich, St Louis, MO, USA) after 4 h of food and water deprivation. After sacrificing all the mice, blood samples were collected in a light-avoiding environment. The serum, diluted 10-fold with ddH_2_O, was excited at 485 nm and the value was measured at 528 nm using a fluorometer (BioTek, Winooski, VT, USA). The intestinal tissue was measured for length and bathed in a 10% formaldehyde solution at room temperature. Protein was extracted from the colon tissue using a protein extraction buffer containing a protease inhibitor. The simplified experimental procedure is shown in Figure 1B.

#### 2.4.1. The Determination of Occult Stool Blood and Scoring the Disease Activity Index (DAI) in DSS-Colitis Mice

The occult blood in stool was measured using the S-Y Feces Occult Blood Reagents I and II (Shin-Ying Medical Instruments, Taipei, Taiwan) with all mouse fecal samples, following the manufacturer’s protocol. The disease activity index was scored based on body weight changes, type of feces, and the level of occult blood in stool, as described by Xia et al. [23] and shown in Table 1.

#### 2.4.2. Hematoxylin and Eosin (HE) Stain for Histological Evaluation

Small intestinal tissue was washed with PBS, formalin-fixed, and then stained with hematoxylin and eosin. A professional veterinarian assessed the histological score according to the scoring criteria shown in Table 2.

#### 2.4.3. Detection of the Mucosal Barrier Protein Expression

Colon tissue was lysed in T-PERTM tissue extraction buffer (Thermo, Waltham, MA, USA) containing a protease inhibitor (Abcam, ab270055), centrifuged at 13,500 rpm for 10 min, boiled at 90 °C for 10 min with 6X Laemmli SDS Sample Buffer (SBN06-15, Visual Protein, Taipei, Taiwan), and analyzed using reducing 10% sodium dodecyl sulfate polyacrylamide gel electrophoresis (SDS-PAGE). Western blotting was performed using antibodies against occludin (Abcam, ab15098, 1:1000), myeloperoxidase (MPO, Abcam, ab208670, 1:1000), ZO-1 (Abcam, ab276131, 1:1000), Mucin 2 (MUC2, ab272692, 1:1000), and β-actin (Abcam, ab16039, 1:1000). The Western blot analysis was quantitative and was performed using ImageJ, with the relative intensities of the target proteins normalized to β-actin shown.

### 2.5. Specific Cecal Bacteria Determination

The DNA from the cecal contents of mice was extracted, and the genera *Lactobacillus*, *Enterobacteriaceae*, *Bifidobacterium*, and *Clostridium* were quantified through quantitative polymerase chain reaction (qPCR) using specific primers with an ABI machine. The specific primers for *Bifidobacterium* are g-Bifid-F: 5′-CTCCTGGAAACGGGTGG-3′ and g-Bifid-R: 5′-GGTGTTCTTCCCGATATCTACA-3′ [25]; for *Enterobacteriaceae*, En-Isu-3F: 5′-TGCCGTAACTTCGGGAGAAGGCA-3′ and En-Isu-3R: 5′-TCAAGGACCAGTGTTCAGTGTC-3′ [26]; for *Lactobacillus*, g-Lact (Rin)-1: 5′-AGCAGTAGGGAATCTTCCA-3′ and g-Lact (Rin)-2: 5′-CACCGCTACACATGGAG-3′ [27]; and for *Clostridium*, CIPER-F: 5′-AGATGGCATCATCATTCAAC-3′ and CIPER-R: 5′-GCAAGGGATGTCAAGTGT-3′ [28].

### 2.6. The Concentration of Cytokines

The concentrations of cytokines, including tumor necrosis factor-alpha (TNF-α), interleukin (IL)-1β, IL-5, IL-6, and IL-10 were measured using mouse cytokine ELISA kits (R&D Systems, Inc., McKinley, MN, USA) according to the manufacturer’s instructions.

### 2.7. Statistical Analysis

The data, presented as the mean ± standard deviation of triplicates, were expressed as percentages relative to control values. Statistical comparisons were conducted using Tukey’s test within the GraphPad Prism 6.0 software. Significance levels were denoted as * *p* < 0.05, ** *p* < 0.01, and *** *p* < 0.001 compared to the positive control and ^#^ *p* < 0.05, ^##^ *p* < 0.01, and ^###^ *p* < 0.001 compared to the negative control.

## 3. Results

### 3.1. Effects of Lactobacillus acidophilus TW01 on Physiological Responses in PM_2.5_-Induced Lung Injury in Mice

OVA sensitization followed by PM_2.5_ exposure resulted in a significant decrease in body weight compared to the control group (Ctrl) starting from day 8 (Figure 2A, *p* < 0.001). This weight loss was observed in both the negative control (NC) and high-dose (*L. acidophilus* TW01) (HD) groups. By day 15, all OVA/PM_2.5_-exposed groups exhibited significantly lower body weights than the Ctrl group until day 22, after which body weight gradually recovered (Figure 2A, *p* < 0.001). Additionally, OVA sensitization and PM_2.5_ exposure significantly increased the spleen-to-body weight ratio compared to the Ctrl group (Figure 2B, *p* < 0.0001). The HD group exhibited a trend toward reduced splenomegaly (*p* = 0.0521), though this did not reach statistical significance.

In terms of immunoregulatory effects, OVA sensitization and PM_2.5_ exposure significantly increased total serum IgG (Figure 3A, *p* < 0.05) and showed a trend toward increased IgG1 levels (Figure 3B, *p* > 0.05) in the NC group compared to the Ctrl group. The HD group showed a trend reduction in IgG1 (Figure 3B) levels, as well as in the IgG1/IgG2a ratio (Figure 3D) and IgE levels (Figure 3E). The IgG2a level showed no significant change (Figure 3C).

Analysis of bronchoalveolar lavage fluid (BALF) revealed that the NC group had increased levels of the pro-inflammatory cytokines TNF-α and IL-6 compared to the Ctrl group (Figure 4A,B). The expression of IL-10 showed no significant change (Figure 4C). IL-5 levels were also elevated in the NC group (Figure 4D). Both low-dose (LD) and HD *L. acidophilus* TW01 treatments reduced TNF-α and IL-6 levels, with the HD group showing statistically significant reductions in all three cytokines (TNF-α, IL-6, and IL-5, *p* < 0.05) (Figure 4A,B,D). The HD group showed a trend toward reduced total cell counts in BALF compared to the NC group (Figure 4E, *p* = 0.058).

Western blot analysis indicated that OVA sensitization and PM_2.5_ exposure resulted in increased expression of phosphorylated Smad3, total Smad3, and the phosphorylated Smad3/total Smad3 ratio in lung tissue (Figure 5A,B). Both LD and HD *L. acidophilus* TW01 treatments tended to downregulate these markers, particularly phosphorylated Smad3, with significant reductions observed in the HD group (Figure 5B, *p* < 0.05). The level of IL-1β showed no significant change (Figure 5C).

Furthermore, compared to the Ctrl group, the NC group exhibited a trend toward decreased abundance of *Clostridium* and *Bifidobacterium* (Figure 6A,C), with no significant change in *Lactobacillus* levels (Figure 6B). Both LD and HD *L. acidophilus* TW01 treatments significantly increased the abundance of *Lactobacillus* and *Bifidobacterium,* with the LD group showing a particularly significant increase in *Lactobacillus* (*p* < 0.05).

### 3.2. Lactobacillus acidophilus TW01 Ameliorates DSS-Induced Colitis by Modulating Cytokines, Improving Gut Barrier Integrity, and Upregulating Bifidobacterium

Given the association between chronic PM_2.5_ exposure and inflammatory bowel disease (IBD), we further investigated the protective effects of *L. acidophilus* TW01 in a DSS-induced colitis model in vivo. There were no significant differences in body weight among the groups (Figure 7A). However, the disease activity index (DAI) score (Figure 7B) increased significantly in the negative control (NC) group from day 9 onward (*p* < 0.05) compared to the control (Ctrl) group. In contrast, both the low-dose (TW01 LD) and high-dose (TW01 HD) *L. acidophilus* TW01 groups exhibited significantly lower DAI scores than the NC group from days 9 to 12 (*p* < 0.05), with the LD group maintaining a significant difference until day 13 (*p* < 0.05). The commercial probiotic (P) group displayed DAI scores comparable to those of the NC group. DSS also significantly increased the spleen-to-body weight ratio compared to the Ctrl group (*p* < 0.01) (Figure 7C), an effect that was not significantly altered by either *L. acidophilus* TW01 dosage or the P group (Figure 7C).

DSS treatment significantly reduced colon length compared to the Ctrl group (Figure 8A,B, *p* < 0.001), but this reduction was not significantly mitigated by *L. acidophilus* TW01 treatment (Figure 8B). Histological analysis (Figure 8C–E, Appendix A) revealed significant increases in inflammation scores in the NC group (*p* < 0.001) compared to the Ctrl group, while both TW01 LD and TW01 HD groups showed significantly lower inflammation scores than the NC group (*p* < 0.05), except for submucosal edema in the TW01 LD group.

Importantly, treatment with *L. acidophilus* TW01 (both TW01 LD and TW01 HD groups) significantly reduced DSS-induced gut permeability (measured by FITC-dextran) compared to the NC group (Figure 8F, *p* < 0.001). Consistently, all treatment groups (P, TW01 LD, and TW01 HD) exhibited significantly lower IL-1β levels in large intestinal tissue compared to the NC group (Figure 9C, *p* < 0.001). However, there were no significant differences among the groups in TNF-α and IL-10 levels (Figure 9A,B).

Additionally, DSS treatment significantly increased myeloperoxidase (MPO) expression (Figure 10A,B, *p* < 0.001) compared to the Ctrl group, an effect that was not significantly altered by *L. acidophilus* TW01 treatment (Figure 10B). Regarding tight junction proteins, only the P group significantly increased ZO-1 expression compared to the Ctrl group (Figure 10C, *p* < 0.05); neither the TW01 LD nor TW01 HD group significantly affected ZO-1 expression. MUC2 is the marker for the colon protective gel barrier. However, each treatment group showed no difference compared to the Ctrl group (Figure 10C). DSS treatment significantly decreased occludin-1 expression (Figure 10C, *p* < 0.05), and neither TW01 LD nor TW01 HD group significantly altered occludin-1 expression compared to the NC.

In terms of microbial levels, DSS treatment did not significantly affect the abundance of *Enterobacteriaceae*, *Clostridium*, or *Lactobacillus* (Figure 11A–C). However, both the low-dose and high-dose *L. acidophilus* TW01 treatments significantly increased *Bifidobacterium* populations compared to both Ctrl and NC groups (Figure 11D, *p* < 0.001).

## 4. Discussion

This study explored the protective effects of *L. acidophilus* TW01 against PM_2.5_-induced lung injury and DSS colitis in mice, highlighting the intricate interplay between lung and gut health. Treatment with *L. acidophilus* TW01 reduced PM_2.5_-induced BALF total cells, evidenced by decreased levels of pro-inflammatory cytokines TNF-α and IL-6 and allergic IL-5. Furthermore, it mitigated fibrotic responses, as indicated by lower levels of TGF-β1 and phosphorylated Smad3, alongside modulation of serum immunoglobulin levels, suggesting a comprehensive immunomodulatory effect. Concurrently, *L. acidophilus* TW01 improved gut barrier function and altered gut microbiota composition in both DSS and PM_2.5_ models, notably increasing populations of beneficial bacteria such as *Bifidobacterium* and *Lactobacillus*. This observation hints at a potential mechanistic link between gut health and lung protection, emphasizing the relevance of the gut–lung axis in understanding the systemic effects of environmental stressors.

Understanding the outcomes of this investigation requires careful consideration of the varying experimental protocols employed. PM_2.5_ is a complex, heterogeneous mixture derived from diverse sources, each component possessing distinct chemical and biological toxicities. This variability contributes to the differential impacts observed with PM_2.5_ exposure [29]. For instance, the method of PM_2.5_ administration, such as intranasal administration versus intranasal instillation, along with the duration of exposure, can significantly influence resultant health outcomes. As demonstrated in a cohort study, prolonged exposure to elevated levels of PM_2.5_ correlates with increased risk of chronic respiratory diseases [30].

Despite these challenges, numerous studies have consistently reported a mixed Th2/Th1/Th17 immune response as a response to PM_2.5_, rather than a purely Th2-dominant subtype. This complexity suggests a potentially severe form of asthma, further underscoring the multifaceted effects of PM_2.5_ [31,32,33]. It is noteworthy that, in our study, we observed no significant lung damage following OVA sensitization and PM_2.5_ intranasal instillation, potentially due to the low sulfate content of the PM_2.5_ components. Previous research has indicated that sulfate can exacerbate lung injury and inflammation, given its ability to penetrate the body and induce vascular endothelial cell damage [34,35,36].

The protective mechanisms of *L. acidophilus* TW01 further illuminate its anti-inflammatory properties within the context of PM_2.5_ exposure. Activation of the immune system from PM_2.5_ is characterized by the mobilization of immune cells, such as macrophages and T-helper cells [37]. These activated immune cells promptly release pro-inflammatory cytokines like TNF-α and IL-6, which are central to the inflammatory process [38]. Importantly, IL-6, in conjunction with IL-5, promotes the differentiation and proliferation of B-cells, enhancing the production of antibodies, including IgG and its subtypes IgG1 and IgG2 [39,40]. Furthermore, IL-5 is crucial for eosinophil maturation and activation, thereby contributing to allergic responses and regulating IgG production [41]. TNF-α and IL-6 trigger the release of IL-1β within the lung tissue, amplifying the inflammatory response and promoting the production of TGF-β1, a known pro-fibrotic factor. TGF-β1 interacts with its receptors to initiate the phosphorylation of Smad3 (*p*-Smad3), which then complexes with other Smad proteins to translocate to the nucleus and modulate gene expression related to immune regulation, cell growth, and differentiation, ultimately facilitating fibrotic processes [42,43,44,45].

*Lactobacillus acidophilus* TW01 exerts a protective effect against PM_2.5_-induced lung injury by downregulating inflammatory markers, specifically TNF-α and IL-6, and by mitigating allergic reactions mediated by IL-5. Additionally, this probiotic reduces overall antibody production, including IgG and its subtypes (IgG1 and IgG2). Furthermore, *L. acidophilus* TW01 lowers IL-1β levels in lung tissue, thereby inhibiting the amplification of the inflammatory response and the activation of the TGF-β1/Smad signaling pathway. These findings suggest that *L. acidophilus* TW01 holds promise as a therapeutic agent to counteract the adverse effects of PM_2.5_ exposure, significantly contributing to immune balance and inflammation reduction.

Various probiotic strains have been documented to alleviate PM_2.5_-induced exacerbations of allergic airway inflammation. Improvements noted include reduced airway hyperresponsiveness [31,33], decreased infiltration of inflammatory cells [32], and a shift towards a more balanced Th1/Th2 immune response [31,32]. These observations imply that probiotics may modulate the inflammatory response broadly, transcending strain-specific effects and highlighting their potential as a universal therapeutic strategy for PM_2.5_-exacerbated asthma. Our study complements these findings by not only demonstrating that *L. acidophilus* TW01 exhibits immunomodulatory properties but also emphasizing its gut-protective effects.

Environmental factors have increasingly been recognized as significant contributors to the pathogenesis of inflammatory bowel disease (IBD). Among these, air pollution has emerged as a critical risk factor, with studies indicating its role in exacerbating IBD symptoms and influencing disease onset [11,12,13]. Specifically, chronic exposure to particulate matter (PM_2.5_) and nitrogen dioxide (NO_2_) has been associated with epigenetic changes that predispose individuals to IBD. These changes include alterations in DNA methylation at key loci, such as the chemokine receptor CXCR2, which is crucial for immune cell trafficking, and various regions within the MHC class III genes that are essential for immune function [11]. Such epigenetic modifications can disrupt regulatory mechanisms, leading to the chronic inflammation typical of IBD.

In gut-protective contexts, *L. acidophilus* TW01 has shown promise in mitigating the severity of colitis induced by DSS, a well-established model for IBD. The administration of *L. acidophilus* TW01 led to a significant reduction in inflammatory markers, particularly IL-1β, which is integral in driving inflammatory responses and tissue damage in IBD. This downregulation indicates that *L. acidophilus* TW01 likely exerts its protective effects through modulation of inflammatory pathways, fostering an environment more conducive to healing. Additionally, this probiotic strain appears to enhance gut barrier integrity, a critical factor for maintaining intestinal homeostasis and preventing the translocation of harmful bacteria and toxins that could worsen inflammation. Improvements in gut barrier function may be mediated through mechanisms such as the production of short-chain fatty acids and other metabolites that promote epithelial cell health and proliferation [46,47,48].

However, while *L. acidophilus* TW01 demonstrated benefits in reducing inflammation and supporting gut barrier integrity, specific tight junction proteins, such as zonula occludens-1 (ZO-1), occludin, and mucin-2 (MUC-2), did not demonstrate enhancement following treatment. In our previous study, *L. acidophilus* TW01 was shown to enhance the migration of Caco-2 cells and demonstrated potential in healing damaged intestinal cells [20]. This suggests a protective effect on the gut barrier, leading to reduced intestinal permeability in DSS-induced colitis, although it does not promote tight junction protein expression. This finding suggests a potential limitation in the probiotic’s efficacy, implying that while it reduces systemic inflammation, additional strategies or combined therapies may be necessary to effectively strengthen tight junction integrity and enhance mucus production. Additionally, while our study demonstrates correlations between *L. acidophilus* TW01 treatment and reduced inflammation, the specific causal mechanisms remain to be further explored.

The bidirectional gut–lung axis is a crucial concept for understanding how environmental pollutants affect respiratory health [4]. Dysbiosis of gut microbiota can significantly alter immune responses, mediating the systemic effects associated with particulate matter (PM) exposure. The significance of the gut–lung axis has become increasingly clear with the identification of various components and metabolites derived from gut microbes, particularly short-chain fatty acids (SCFAs) [16]. These metabolites play a crucial role in modulating the immune system. Research indicates that the gut microbiota ferments undigested soluble dietary fibers to produce SCFAs [18], which can directly and indirectly influence the function of various cell types, including epithelial cells and both innate and adaptive immune cells [49]. As key signaling molecules, SCFAs are instrumental in limiting inflammation and promoting protective immune responses both within the gut and throughout the body [50].

Our research supports this model by demonstrating the protective effects of *L. acidophilus* TW01. Treatment with *L. acidophilus* TW01 not only mitigates PM_2.5_-induced lung injury, as evidenced by decreased pro-inflammatory cytokines in bronchoalveolar lavage fluid (BALF) and improved lung function, but also enhances gut health by increasing the abundance of beneficial bacteria such as *Bifidobacterium* and *Lactobacillus* and improving gut barrier integrity. This dual modulation of lung and gut health indicators strongly suggests a functional interaction within the gut–lung axis, corroborating findings from Dai et al. [51,52].

## 5. Conclusions

In conclusion, this study highlights the protective effects of *L. acidophilus* TW01 against PM_2.5_-induced lung injury, emphasizing the complex interplay between lung and gut health. Treatment with *L. acidophilus* TW01 reduced inflammatory markers, specifically TNF-α and IL-6, and mitigated allergic reactions mediated by IL-5. This reduction leads to decreased serum levels of IgG and its subtypes (IgG1 and IgG2), as well as lower IL-1β levels in lung tissue, thereby inhibiting the inflammatory response and the activation of the TGF-β1/Smad signaling pathway. Furthermore, *L. acidophilus* TW01 improved gut barrier function and increased beneficial gut microbiota, suggesting a potential connection between gut health and lung protection.

This dual effect highlights the relevance of the gut–lung axis in mediating the systemic effects of environmental pollutants. The novelty of this study lies in demonstrating this connection and the multi-targeted protective effects of *L. acidophilus* TW01 against PM_2.5_-induced lung damage, a significant public health issue. However, limitations in this study raise questions about the mechanisms of action of *L. acidophilus* TW01 and suggest that additional strategies or combined therapies may be necessary to fully reinforce gut barrier function and mucosal health. Furthermore, extending the duration of PM_2.5_ exposure is needed to better understand its impact on PM_2.5_-induced IBD in mice, optimizing its application and elucidating the extent of its protective mechanisms.

## Figures and Tables

**Figure 1 nutrients-17-00831-f001:**
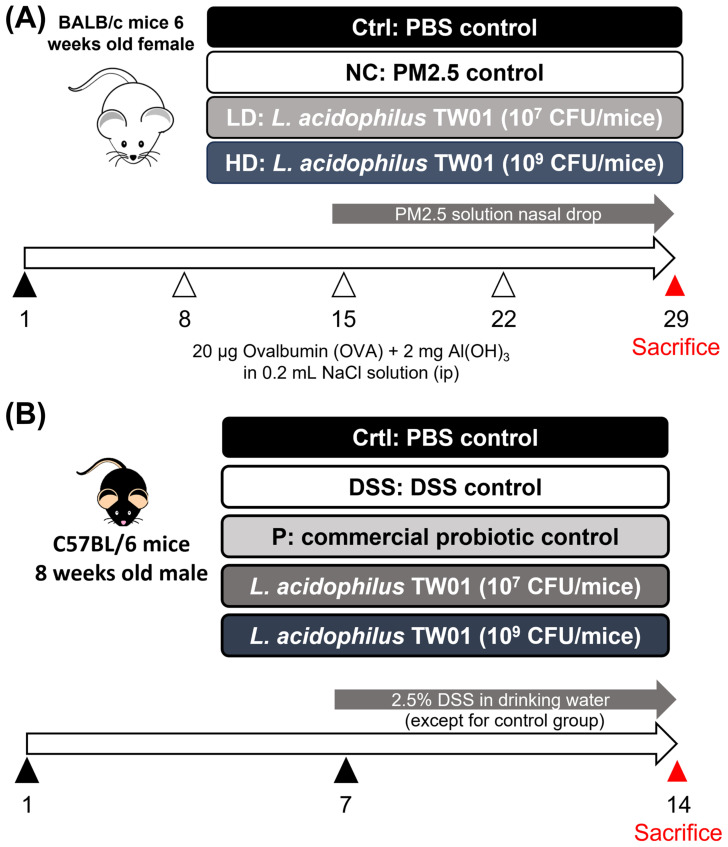
Schematic diagram of the experimental design for (**A**) PM_2.5_ lung-injury mouse model and (**B**) DSS-colitis mouse model.

**Figure 2 nutrients-17-00831-f002:**
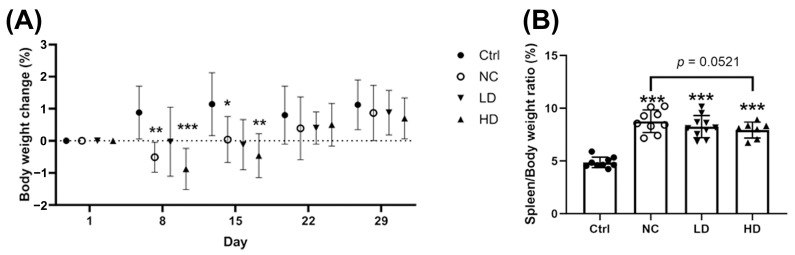
Effects of *L. acidophilus* TW01 in the PM_2.5_ mouse model. (**A**) Changes in body weight. (**B**) Spleen-to-body weight ratio in PM_2.5_-exposed mice. Ctrl: PBS control; NC: OVA + PM_2.5_ control; LD: low-dose *L. acidophilus* TW01; HD: high-dose *L. acidophilus* TW01. Data are presented as mean ± SD (n = 7–10). Significance vs. control, * *p* < 0.05, ** *p* < 0.01, and *** *p* < 0.001.

**Figure 3 nutrients-17-00831-f003:**
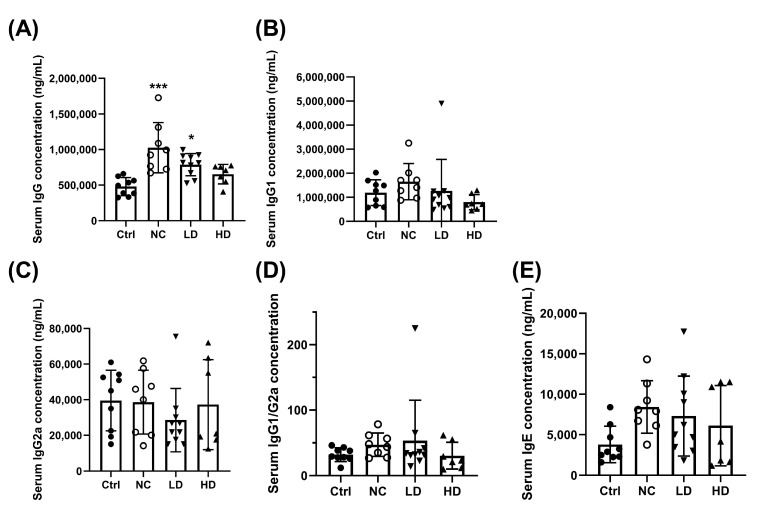
Expression of (**A**) total IgG, (**B**) IgG1, (**C**) IgG2, (**D**) IgG1/G2a, and (**E**) total IgE in PM_2.5_ mice serum. Ctrl: PBS control; NC: OVA + PM_2.5_ control; LD: low-dose *L. acidophilus* TW01; HD: high-dose *L. acidophilus* TW01. Data are presented as mean ± SD (n = 7–10). Statistically significant difference between the Control and treated groups at * *p* < 0.05 and *** *p* < 0.001.

**Figure 4 nutrients-17-00831-f004:**
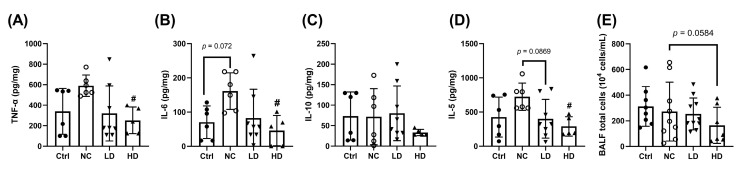
(**A**) TNF-α, (**B**) IL-6, (**C**) IL-10, (**D**) IL-5, and (**E**) total cell counts in BALF of PM_2.5_ mice. Ctrl: PBS control; NC: OVA+PM_2.5_ control; LD: low-dose *L. acidophilus* TW01; HD: high-dose *L. acidophilus* TW01. Data are presented as mean ± SD (n = 7–10). Statistically significant difference between the NC and treated groups at ^#^ *p* < 0.05.

**Figure 5 nutrients-17-00831-f005:**
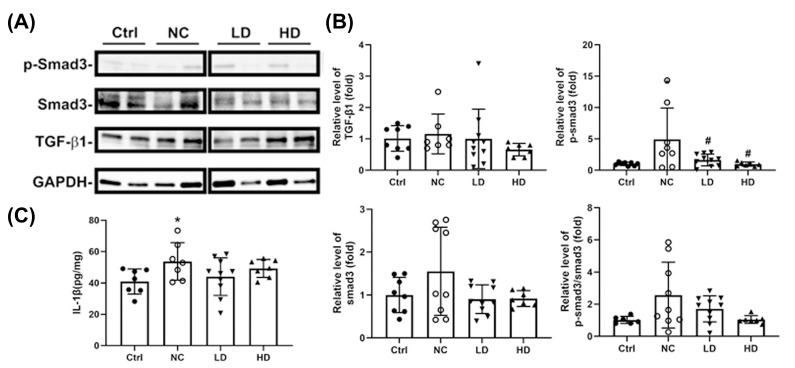
Western blot analysis of *L. acidophilus* TW01’s effect on protein expression in lung tissue of PM_2.5_-exposed mice. (**A**) Western blot analysis. (**B**) Protein expression levels of fibrosis markers as determined by western blot. (**C**) Levels of IL-1β. Data are presented as mean ± SEM; n = 6–10. Ctrl: PBS control; NC: OVA + PM_2.5_ control; LD: low-dose *L. acidophilus* TW01; HD: high-dose *L. acidophilus* TW01. Statistically significant difference between the Control and treated groups at * *p* < 0.05. Statistically significant difference between the NC and treated groups at ^#^ *p* < 0.05.

**Figure 6 nutrients-17-00831-f006:**
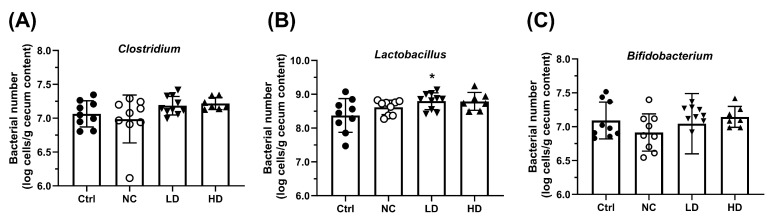
Effect of *L. acidophilus* TW01 on the abundance of (**A**) *Clostridium*, (**B**) *Lactobacillus*, and (**C**) *Bifidobacterium* in the colon contents of PM_2.5_-exposed mice. Ctrl: PBS control; NC: OVA + PM_2.5_ control; LD: low-dose *L. acidophilus* TW01; HD: high-dose *L. acidophilus* TW01. Data are presented as mean ± SD (n = 6–10). Significance vs. control, * *p* < 0.05.

**Figure 7 nutrients-17-00831-f007:**
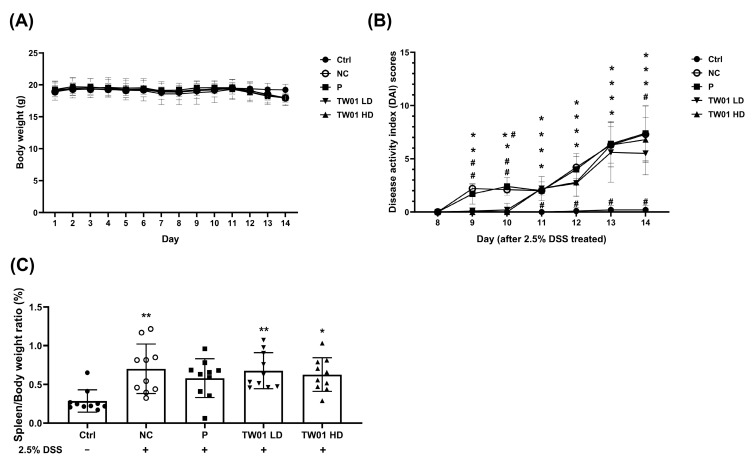
Effects of *L. acidophilus* TW01 on the DSS colitis mouse model. (**A**) Changes in body weight. (**B**) Disease Activity Index (DAI) scores. (**C**) Spleen-to-body weight ratio of DSS-treated mice. Ctrl: control group, DSS: DSS control group, P: commercial probiotics group, TW01 LD: low-dose *L. acidophilus* TW01 group, TW01 HD: high-dose *L. acidophilus* TW01 group. Data are presented as mean ± SD (n = 9–10). Significance vs. control, * *p* < 0.05, and ** *p* < 0.01. Significance vs. DSS, ^#^ *p* < 0.05.

**Figure 8 nutrients-17-00831-f008:**
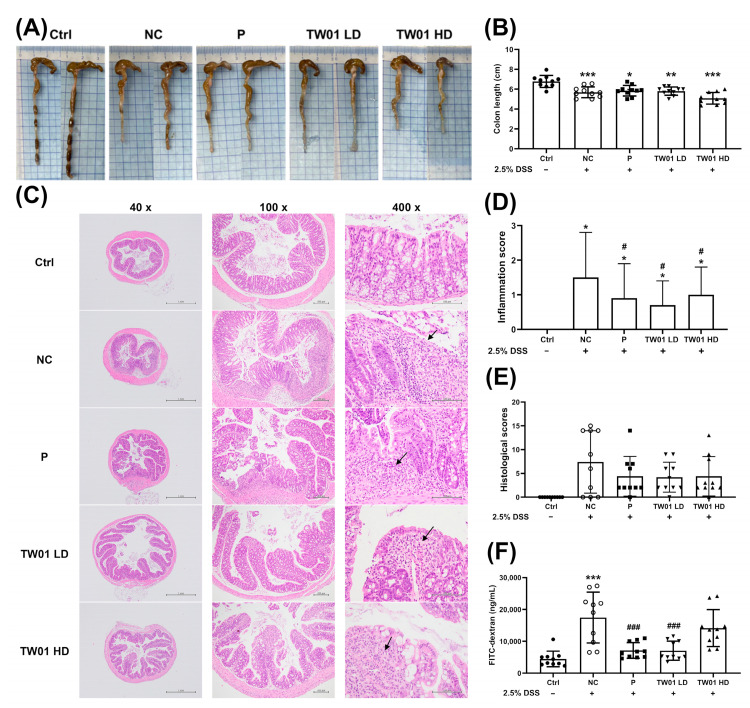
Pathological effects of *L. acidophilus* TW01 in the DSS colitis mouse model. (**A**) Photograph of the colon. (**B**) Statistical analysis of colon length. (**C**) Histopathological examination of colon sections (scale bars = 1 mm, 200 μm, and 100 μm). Inflammatory cell infiltration in the mucosal layer of the colon is indicated by black arrows. (**D**) Inflammation scores for the colon. (**E**) Total histopathology scores in the colon. (**F**) Assessment of gut permeability in DSS-treated mice. Ctrl: control group, DSS: DSS control group, P: commercial probiotics group, TW01 LD: low-dose *L. acidophilus* TW01 group, TW01 HD: high-dose *L. acidophilus* TW01 group. Data are presented as mean ± SD (n = 9–10). Significance vs. control, * *p* < 0.05, ** *p* < 0.01, and *** *p* < 0.001. Significance vs. DSS, ^#^ *p* < 0.05, and ^###^ *p* < 0.001.

**Figure 9 nutrients-17-00831-f009:**
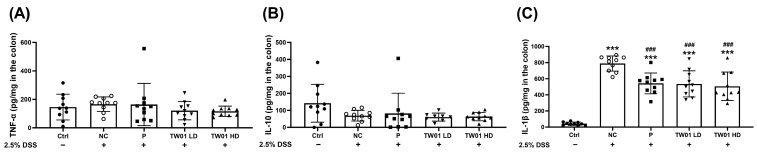
Effect of *L. acidophilus* TW01 on (**A**) TNF-α, (**B**) IL-10, and (**C**) IL-1β levels of DSS mice large intestinal tissue. Ctrl: control group, DSS: DSS control group, P: commercial probiotics group, TW01 LD: low-dose *L. acidophilus* TW01 group, TW01 HD: high-dose *L. acidophilus* TW01 group. Data are presented as mean ± SD (n = 9–10). Significance vs. control, *** *p* < 0.001. Significance vs. DSS, ^###^ *p* < 0.001.

**Figure 10 nutrients-17-00831-f010:**
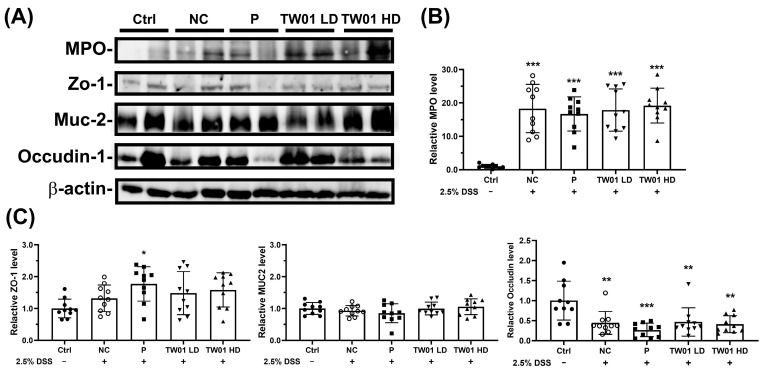
Effect of *L. acidophilus* TW01 on large intestinal tissue of DSS mice. (**A**) The western blot results of protein expression. (**B**) The statistics analysis of myeloperoxidase (MPO) expression in large intestinal tissue. (**C**) The statistics analysis of tight-junction protein level. Ctrl: control group, DSS: DSS control group, P: commercial probiotics group, TW01 LD: low-dose *L. acidophilus* TW01 group, TW01 HD: high-dose *L. acidophilus* TW01 group. Data are presented as mean ± SD (n = 9–10). Significance vs. control, * *p* < 0.05, ** *p* < 0.01 and *** *p* < 0.001.

**Figure 11 nutrients-17-00831-f011:**
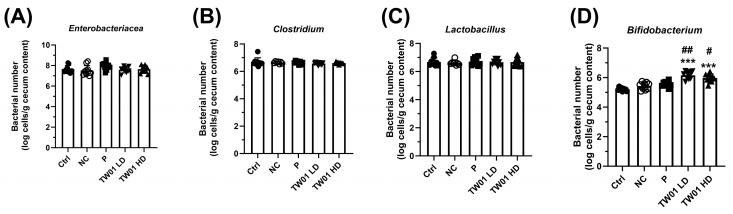
Effect of *L. acidophilus* TW01 on the microbiota of DSS mice colon content. (**A**) *Enterobacteriaceae*, (**B**) *Clostridium*, (**C**) *Lactobacillus*, and (**D**) *Bifidobacterium*. Ctrl: control group, NC: DSS control group, P: commercial probiotics group, TW01 LD: low-dose *L. acidophilus* TW01 group, TW01 HD: high-dose *L. acidophilus* TW01 group. Data are presented as mean ± SD (n = 9–10). Significance vs. control, *** *p* < 0.001. Significance vs. DSS; ^#^
*p* < 0.05 and ^##^
*p* < 0.01.

**Table 1 nutrients-17-00831-t001:** The scoring system of disease activity index (DAI).

Score	Weight Loss (%)	Stool Consistency	Blood in Stool
0	0	Normal	Negative (no bleeding)
1	1–5	-	No blood trance on fecal but slightly positive
2	5–10	Loose stool	No blood trance on fecal but positive
3	10–15	-	Blood fecal with moderate positive
4	>15	Diarrhea	Blood fecal with strong positive

Normal stool = well-formed pellets; loose = pasty stool that does not stick to the anus; diarrhea = liquid stool that sticks to the anus.

**Table 2 nutrients-17-00831-t002:** The histological scoring system.

Score	Scribe
1	Lost, crypt
2	Regeneration, crypt
3	Edema, submucosa
4	Inflammation, mononuclear cells
5	Ulcer, with fibroblast cell infiltration

Severity of lesions was graded according to the methods described by Shackelford et al. (2002) [24]. Degree of lesions was graded from one to five depending on severity: 1 = minimal (<1%); 2 = slight (1–25%); 3 = moderate (26–50%); 4 = moderate/severe (51–75%); 5 = severe/high (76–100%).

## Data Availability

Data are contained within the article.

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
