# Peer review of "Lactobacillus acidophilus TW01 Mitigates PM2.5-Induced Lung Injury and Improves Gut Health in Mice"

_nutrients, 2025, doi:10.3390/nu17050831_

Round 1

Reviewer 1 Report

Comments and Suggestions for Authors

The checklist document received via email provides an appropriately detailed description of the animal experiment process. The research complies with the 3 R principles.

According to the study’s results, the probiotic reduced inflammatory marker levels and improved gut barrier function. Additionally, it increased the proportion of beneficial bacteria, such as Bifidobacterium and Lactobacillus.

Some tight junction proteins did not show any changes, and the reason and significance of this remain unclear. Why did the tight junction proteins not exhibit changes? Is there an explanation for this? How long might the protective effect of the probiotic last, and what could be the optimal dosage? As a possible addition, the paper could also discuss how probiotic administration affects the gut-lung axis in humans and what other mechanisms might contribute to this process.

The research contributes to the understanding of the therapeutic applications of probiotics. The article is thorough, well-written, and scientifically sound, with relevant references. The plagiarism index is appropriate, and the study effectively illustrates the role of the gut-lung axis in mitigating inflammatory responses induced by PM 2.5.

Author Response

Comments from Reviewer 1

  1. The checklist document received via email provides an appropriately detailed description of the animal experiment process. The research complies with the 3 R principles. According to the study’s results, the probiotic reduced inflammatory marker levels and improved gut barrier function. Additionally, it increased the proportion of beneficial bacteria, such as Bifidobacterium and Lactobacillus.

Response: Thank you for your comments.

  1. Some tight junction proteins did not show any changes, and the reason and significance of this remain unclear. Why did the tight junction proteins not exhibit changes? Is there an explanation for this?

Response: Thank you for your comments. The outcome of DSS-induced colitis can be influenced by factors such as animal species, strain specificity, intestinal microflora, and housing conditions (Kim et al., 2012). In our previous study, L. acidophilus TW01 was shown to enhance the migration of Caco-2 cells and demonstrated potential in healing damaged intestinal cells (Luo et al., 2023). This implies a protective effect on the gut barrier, resulting in reduced intestinal permeability in DSS-induced colitis, although it does not enhance tight junction protein expression. We have included this explanation in the discussion section as follows:

“In our previous study, L. acidophilus TW01 was shown to enhance the migration of Caco-2 cells and demonstrated potential in healing damaged intestinal cells (Luo et al., 2023). This suggests a protective effect on the gut barrier, leading to reduced intestinal permeability in DSS-induced colitis, although it does not promote tight junction protein expression.” Please check the revised manuscript in lines 400-403.

References:

Kim J. J, Shajib M. S., Manocha M. M., Khan W. I. Investigating intestinal inflammation in DSS-induced model of IBD. J Vis Exp. 2012 Feb 1;(60):3678. doi: 10.3791/3678.

Luo, S., Chen, M. Systematic Investigation of the Effect of Lactobacillus acidophilus TW01 on Potential Prevention of Particulate Matter (PM)2.5-Induced Damage Using a Novel In Vitro Platform. Foods 202312, 3278. doi: /10.3390/foods12173278.

  1. How long might the protective effect of the probiotic last, and what could be the optimal dosage?

Response: Thank you for your comments. In comparing the results from the two groups of mice, the high-dose group outperformed the low-dose group in the PM₂.₅-induced lung injury model. In contrast, both the high-dose and low-dose groups exhibited similar effects in the DSS-induced model. Therefore, a dosage of 1 × 10⁹ CFU/mouse appears to be the more favorable option. However, our current study did not assess the long-term protective effects of the probiotics, which will require further experimentation for validation. We have revised the conclusion section as follows:

“Furthermore, extending the duration of PM₂.₅ exposure is needed to better understand its impact on PM₂.₅-induced IBD in mice, optimizing its application and elucidating the extent of its protective mechanisms.”  Please check the revised manuscript in lines 446-448.

  1. As a possible addition, the paper could also discuss how probiotic administration affects the gut-lung axis in humans and what other mechanisms might contribute to this process.

Response: Thank you for your comments. We have added the discussion as follows:

“The significance of the gut-lung axis has become increasingly clear with the identification of various components and metabolites derived from gut microbes, particularly short-chain fatty acids (SCFAs) (Dang et al., 2019). These metabolites play a crucial role in modulating the immune system. Research indicates that the gut microbiota ferments undigested soluble dietary fibers to produce SCFAs (Keulers et al., 2022), which can directly and indirectly influence the function of various cell types, including epithelial cells and both innate and adaptive immune cells (Sharon et al., 2014). As key signaling molecules, SCFAs are instrumental in limiting inflammation and promoting protective immune responses both within the gut and throughout the body (Garrett et al., 2010).” Please check the revised manuscript in lines 413-421.

References

Dang A. T., Marsland B. J., Microbes, metabolites, and the gut-lung axis. Mucosal Immunol, 2019. 12(4): p. 843-850. doi: 10.1038/s41385-019-0160-6.

Keulers L., Dehghani A., Knippels L., Garssen J., Papadopoulos N., Folkerts G., Braber S., Bergenhenegouwen J. V.,Prebiotics, and synbiotics to prevent or combat air pollution consequences: The gut-lung axis. Environ Pollut, 2022. 302: 119066. doi: 10.1016/j.envpol.2022.119066.

Sharon G., Garg N., Debelius J., Knight R., Dorrestein P. C, Mazmanian S. K, Specialized metabolites from the microbiome in health and disease. Cell Metab, 2014. 20(5): p. 719-730. doi: 10.1016/j.cmet.2014.10.016.

Garrett W. S, Gordon J. I, Glimcher L. H, Homeostasis and inflammation in the intestine. Cell, 2010. 140(6): p. 859-70. doi: 10.1016/j.cell.2010.01.023.

  1. The research contributes to the understanding of the therapeutic applications of probiotics. The article is thorough, well-written, and scientifically sound, with relevant references. The plagiarism index is appropriate, and the study effectively illustrates the role of the gut-lung axis in mitigating inflammatory responses induced by PM 2.5.

Response: Thank you for your comments.

Reviewer 2 Report

Comments and Suggestions for Authors

The manuscript presents a study investigating the protective effects of Lactobacillus acidophilus TW01 against PM2.5-induced lung injury and gut health deterioration in mice. The study is well-structured and provides useful insights into the gut-lung axis. However, there are several concerns regarding the clarity of the introduction, methodological rigor, depth of analysis, and discussion. Below are listed my observations:

The introduction fails to explicitly state what aspects of previous research are missing or how this study uniquely contributes to the field. It would benefit from a direct statement outlining the novelty of this study.

Instead of reiterating the general effects of PM2.5, in the introduction, the authors, should directly relate these effects to the specific mechanism investigated (gut-lung axis modulation).

The description of PM2.5 administration is vague. Details on the specific composition of PM2.5, its origin, and standardization methods should be included.

The selection of the probiotic doses (107 and 109 CFU/mL) is based on prior studies or preliminary experiments?

The authors should ensure consistency in figure labeling throughout the manuscript. Each multi-panel figure should have clear sub-labels (A, B, C, etc.) and corresponding explanations in the figure legend. This will improve clarity and make it easier for readers to interpret the data.

While the study shows correlations between L. acidophilus TW01 treatment and reduced inflammation, direct causal mechanisms are not established. The discussion should acknowledge this limitation.

The conclusion section  states that L. acidophilus TW01 has "therapeutic potential" but does not specify what further steps should be taken (e.g., human trials, mechanistic studies). The limitations should be explicitly acknowledged to balance the claims.

Author Response

Comments from Reviewer 2

The manuscript presents a study investigating the protective effects of Lactobacillus acidophilus TW01 against PM2.5-induced lung injury and gut health deterioration in mice. The study is well-structured and provides useful insights into the gut-lung axis. However, there are several concerns regarding the clarity of the introduction, methodological rigor, depth of analysis, and discussion. Below are listed my observations:

  1. The introduction fails to explicitly state what aspects of previous research are missing or how this study uniquely contributes to the field. It would benefit from a direct statement outlining the novelty of this study.

Response: Thank you for your comments. We have revised the introduction section of this manuscript to better highlight the significance of our study.

“In our previous research, we employed an in-vitro screening model that identified Lactobacillus acidophilus TW01 as a promising strain for mitigating oxidative damage, enhancing wound healing in intestinal epithelial cells, and protecting bronchial cells from cigarette smoke extract (Luo, 2023). Building upon this foundation, the current study uniquely addresses the protective effects of L. acidophilus TW01 against PM-induced lung injury in a mouse model, specifically through the gut-lung axis. By utilizing two distinct groups of mice, we demonstrate not only its novel therapeutic potential but also the underlying mechanisms involved. This study fills critical gaps in our understanding of the gut-lung interaction and supports the development of targeted probiotic therapies for respiratory health.” Please check the revised manuscript in lines 56-65.

References:

Luo, S., Chen, M. Systematic Investigation of the Effect of Lactobacillus acidophilus TW01 on Potential Prevention of Particulate Matter (PM)2.5-Induced Damage Using a Novel In Vitro Platform. Foods 202312, 3278. doi: /10.3390/foods12173278.

  1. Instead of reiterating the general effects of PM2.5, in the introduction, the authors, should directly relate these effects to the specific mechanism investigated (gut-lung axis modulation).

Response: Thank you for your comments. We have modified the introduction section of this manuscript as follow:

“The gut-lung axis describes the connection between respiratory and gut health, where the immune system, microbiota, and their metabolites play significant roles. Both the respiratory and gastrointestinal systems regulate inflammatory responses by modulating the secretion of chemokines and cytokines, which in turn influences immune system activity (Dang et al., 2019). Chronic exposure to air pollution has been shown to alter the composition of the microbiota and affect microbial metabolite production, leading to pro-inflammatory responses and disrupted immune homeostasis (Keulers, 2022).” Please check the revised manuscript in lines 45-51.

References

Dang A. T., Marsland B. J., Microbes, metabolites, and the gut-lung axis. Mucosal Immunol, 2019. 12(4): p. 843-850. doi: 10.1038/s41385-019-0160-6.

Keulers L., Dehghani A., Knippels L., Garssen J., Papadopoulos N., Folkerts G., Braber S., Bergenhenegouwen J. V.,Prebiotics, and synbiotics to prevent or combat air pollution consequences: The gut-lung axis. Environ Pollut, 2022. 302: 119066. doi: 10.1016/j.envpol.2022.119066.

  1. The description of PM2.5 administration is vague. Details on the specific composition of PM2.5, its origin, and standardization methods should be included.

Response: Thank you for your comments. We have expanded the description of the PM₂.₅ administration to provide clearer details on its composition and standardization. The revised text is as follows:

 “On Day 14, we administered 1.8 mg/kg body weight of urban dust (SRM 1649b; National Institute of Standards and Technology, U.S. Department of Commerce, Gaithersburg, MD), which is a standardized reference material for particulate matter. The dust was suspended in 10 μL of phosphate-buffered saline (PBS) and delivered via anterior nasal cavity drops daily until Day 29.” Please check the revised manuscript in lines 85-89.

  1. The selection of the probiotic doses (107 and 109 CFU/mL) is based on prior studies or preliminary experiments?

Response: Thank you for your insightful question. The selected probiotic doses of 10⁷ and 10⁹ CFU/mL are indeed based on our prior studies. Specifically, in our previous research, we observed that a concentration of 10⁸ CFU/mL of L. acidophilus TW01 effectively inhibited reactive oxygen species (ROS) production. Conversely, a lower concentration of 10⁶ CFU/mL promoted the migration of Caco-2 cells. These experimental doses correspond to an approximate oral dosage of 10⁷ and 10⁹ CFU/kg body weight per day in mice, assuming an average body weight of 20 g and a gavage volume of 0.2 mL per day. This rationale supports the chosen doses for our current study.

  1. The authors should ensure consistency in figure labeling throughout the manuscript. Each multi-panel figure should have clear sub-labels (A, B, C, etc.) and corresponding explanations in the figure legend. This will improve clarity and make it easier for readers to interpret the data.

Response: Thank you for your valuable feedback. We have addressed the issue of figure labeling by modifying the manuscript to ensure consistency throughout. Each multi-panel figure now includes clear sub-labels (A, B, C, etc.) along with corresponding explanations in the figure legends. Additionally, we have divided the largest figure (originally Figure 6) into two separate figures (now Figures 6 and 7) to enhance clarity. Please check Figure 2, 3, 5, 6, 7 and 8.

  1. While the study shows correlations between acidophilus TW01 treatment and reduced inflammation, direct causal mechanisms are not established. The discussion should acknowledge this limitation.

Response: Thank you for your insightful suggestion. We have acknowledged this limitation in the revised discussion section of the manuscript. We now emphasize that “while our study demonstrates correlations between L. acidophilus TW01 treatment and reduced inflammation, the specific causal mechanisms remain to be further explored.” Please check the revised manuscript in lines 407-409.

  1. The conclusion section states that acidophilus TW01 has "therapeutic potential" but does not specify what further steps should be taken (e.g., human trials, mechanistic studies). The limitations should be explicitly acknowledged to balance the claims.

Response: Thank you for your comments. We have modified the conclusion section of this manuscript as follow:

“Furthermore, extending the duration of PM₂.₅ exposure is needed to better understand its impact on PM₂.₅-induced IBD in mice, optimizing its application and elucidating the ex-tent of its protective mechanisms.” Please check the revised manuscript in lines 446-448.

Reviewer 3 Report

Comments and Suggestions for Authors

The ARRIVE document is not completely filled out correctly - e.g. The sample size is not only the 40 mice mentioned in L72-73 (as stated in the document), but also in L91-92, which makes a total of 90 mice, which is not in line with the 4R concept (The 4 R concept, alternatives are Reduction, Refining, Replacement and Reproduction).

Be sure to check the ARRIVE document again.

It is also not clear why in the expression "PM2.5" 2.5 is not specified as an index

add PM2.5 to the list of short cuts

 The work has an extremely large number of images (9 of them) consisting of at least 3 images, which is extremely difficult to follow. Given that the work on the topic is a continuation of the study published in the magazine Foods 2023, 12, 3278. https://doi.org/10.3390/foods12173278, the results are presented much more systematically.

Considering that it is necessary to separate the results of a smaller number of animals, I certainly suggest a clearer presentation of the results.

Sincerely

Author Response

Comments from Reviewer 3

  1. The ARRIVE document is not completely filled out correctly - e.g. The sample size is not only the 40 mice mentioned in L72-73 (as stated in the document), but also in L91-92, which makes a total of 90 mice, which is not in line with the 4R concept (The 4 R concept, alternatives are Reduction, Refining, Replacement and Reproduction). Be sure to check the ARRIVE document again.

Response: Thank you for your comments. We have revised the ARRIVE document. Please review the updated document.

  1. It is also not clear why in the expression "PM2.5" 2.5 is not specified as an index add PM2.5 to the list of short cuts

Response: Thank you for your comments. We have updated the notation for PM₂.₅ and included it in the list of abbreviations. Please review the abbreviations section.

  1. The work has an extremely large number of images (9 of them) consisting of at least 3 images, which is extremely difficult to follow. Given that the work on the topic is a continuation of the study published in the magazine Foods 2023, 12, 3278. https://doi.org/10.3390/foods12173278, the results are presented much more systematically.

Considering that it is necessary to separate the results of a smaller number of animals, I certainly suggest a clearer presentation of the results.

Response: Thank you for your comments. We have revised the manuscript and updated the figure legends to include the newly added sub-labels. Additionally, we have divided the largest figure (original Figure 6) into two separate figures (Figure 6 and Figure 7). Please review Figures 2, 3, 5, 6, 7, and 8.

Round 2

Reviewer 2 Report

Comments and Suggestions for Authors

The authors have taken the reviewers' suggestions into account and have improved the quality of the manuscript. I recommend the publication of the article.

Reviewer 3 Report

Comments and Suggestions for Authors

The corrections in the work have all been done correctly and now it is clearer what was the primary idea that the authors have now successfully conveyed through the presented results, discussion and conclusions.